

# Recent Baltic Sea Storm Surge Events From A Climate Perspective

Nikolaus Groll[1], Lidia Gaslikova[1], and Ralf Weisse[1]

[1]Institute of Coastal Systems, Helmholtz-Zentrum Hereon, 21502 Geesthacht, Germany

**Correspondence:** Nikolaus Groll (nikolaus.groll@hereon.de)

**Abstract.** Three storm surge events with return periods between 10 and 100 years have occurred in the western Baltic Sea in recent years (2017, 2019 and 2023). While in most cases such surge events are associated with high wind speeds, two of the three events occurred at relatively moderate wind speeds. The events are analysed and decomposed into the contributions from different factors, such as direct atmospheric effects or of prefilling of the Baltic Sea, which can lead to such extreme water levels. A numerical hindcast simulation is used to place the events and their contributing components into a climate perspective. While the absolute water levels were among the highest in recent decades, the individual contributions of the direct atmospheric effects as well as prefilling were not unusual for two of the three events, and it was rather a combination of water level and prefilling that caused such prominent extreme events. Although the perceived increased frequency of the events may indicate a relation to climate change, the individual contributions were within the range of climate variability observed in recent decades.

## 1 Introduction

Storm surges are the primary cause of coastal flooding along the world's low-lying coastlines (Bernier et al., 2024). In the Baltic Sea, where amplitudes of astronomical tides are small everywhere, considerable storm surge height can occur and pose a threat, in particular in the southwest regions (Wolski et al., 2014; Aakjær and Buch, 2022; Hofstede and Hamann, 2022; Kiesel et al., 2024), the Gulf of Finland (Suursaar and Sooäär, 2007; Averkiev and Klevannyy, 2010), the Gulf of Riga (Suursaar and Sooäär, 2007; Männikus et al., 2019; Suursaar et al., 2006) or the Gulf of Bothnia (Averkiev and Klevannyy, 2010).

In recent years, three very severe storm surge events have been observed in southwestern part, mainly affecting the German and the Danish coastlines. The first event occurred on 4-5 January 2017 and caused peak water levels of about 1.60 to 1.80 m above the mean water level (MW) along the German Baltic Sea coast. This event was ranked under the five highest surges since 1950 at most gauges at the German Baltic Sea coast (Liu et al., 2022, https://stormsurge-monitor.eu). The second event occurred on 2 January 2019 and caused a water level close to the highest observed in Warnemünde since 1954 (Liu et al., 2022). Finally, on 20 October 2023, a severe storm surge event caused widespread flooding in cities such as Flensburg and Schleswig (Germany), led to the breaching of at least seven (regional) dikes and caused damages of more than 200 million euros in Schleswig-Holstein (Kiesel et al., 2024).

While all events were characterized by high surges, their meteorological and oceanographic details differed. The event in 2017 occurred during relatively low wind speeds and was referred to as a "silent storm surge" in the literature (She and Nielsen,





2019). Wind speeds during the 2019 event were higher but still not extreme over the south western Baltic. Finally, during the event in 2023 strong easterly winds that persisted for two days and reached peak wind speeds of 102 km/h occurred (Kiesel et al., 2024).

While strong onshore winds are the primary cause for coastal storm surges, there are other factors that may substantially enhance coastal sea levels. Such factors vary from region to region. The Baltic Sea is a semi-enclosed sea that is connected to the North Sea only through the relatively narrow Danish Straits. During periods of the prevailing westerly winds the sea level gradient across the Danish straits increases, leading to higher inflow and higher Baltic Sea water volumes (Samuelsson and Stigebrandt, 1996). Transports across the Danish Straits can reach values of up to about 45 km$^3$/day in both directions, which

corresponds to a sea level change of about 12 cm/day over the entire Baltic Sea (Mohrholz, 2018). Typically, such variations that may lead to a *prefilling* (Lehmann and Post, 2015; Andrée et al., 2023) of the Baltic Sea have timescales of about 10 days or longer (Soomere and Pindsoo, 2016) and may enhance the mean water level before the onset of a storm (Suursaar et al., 2006; Madsen et al., 2015). As a consequence, similar storms may lead to different water levels depending on prefilling.

Atmospheric variability on timescales shorter than about 10 days may lead to a redistribution of water masses within the

Baltic Sea basin (Kulikov et al., 2015) or between the Baltic Proper and the Gulf of Riga (Männikus et al., 2019). Seiches with periods of up to tens of hours and e–folding times of up to 2 days may develop (Leppärante and Myrberg, 2009) the details of which are still debated and are still not fully understood (e.g. Wübber and Krauss, 1979; Jönsson et al., 2008; Otsmann et al., 2001). When favorably coupled with storm surges or in resonance with atmospheric forcing, such oscillations may contribute to very high sea level extremes at the coast (Suursaar et al., 2006; Weisse and Weidemann, 2017; Wolski and Wiśniewski,

45   2020).

The three very severe storm surge events affecting the southwestern Baltic Sea in 2017, 2019, and 2023 occurred at different wind speeds. It is therefore obvious that other factors must have contributed differently to the severity of the storm surges. While the individual events were to some extend discussed in the literature (e.g. Kiesel et al., 2024; She and Nielsen, 2019; Suursaar et al., 2006; Aakjær and Buch, 2022), a systematic assessment and comparison of the events is lacking.

In this study, we aim at such a comparison from a climate perspective. Using data from a 1958-2023 hydrodynamical hindcast we first decompose the three events and analyse the extent to which different factors such as prefilling have contributed and may account for the observed differences. Based on the multidecadal hindcast we subsequently derive a climatology and assess the extent to which these events have been unusual.

## 2   Data and experiment setup

### 2.1   Numerical Model Data

To analyse water levels from 1958 to 2023, an existing hindcast by Weisse and Weidemann (2017) is used for the period 1958-2011 and extended from 2012 to 2023. For the extension, the same model setup was used to achieve the best possible homogeneity. For the hindcast and its extension, the hydrodynamic model TRIM-NP (Casulli and Stelling, 1998; Kapitza



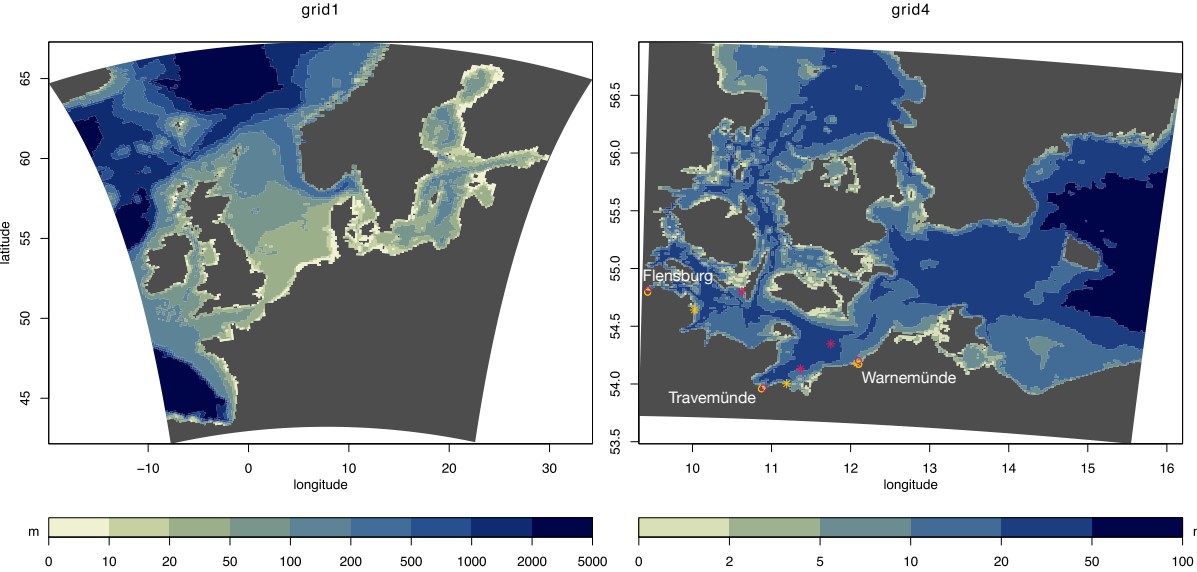

**Figure 1.** Bathymetry for grid 1 and grid 4 of TRIM-NP. Marked are the locations of observed (orange) and modelled (red) water levels (circles) and wind data (asterisk).

and Eppel, 2000; Kapitza, 2008) in barotropic mode was used with a triple nested grid. The coarsest grid covers the northeast

Atlantic with a resolution of 12.8 km and the last nest covers the southwestern Baltic Sea with a resolution of 1.6 km (Figure 1).

Just as with the existing water level hindcast (Weisse and Weidemann, 2017), for the years 2011 to 2023 the model was forced by atmospheric conditions from an extension of the COSMO-CLM atmospheric hindcast simulation (*coastDat2/coastDat3*) (Geyer, 2014). This atmospheric hindcast covers the Euro-Cordex domain (Kotlarski et al., 2014), has a spatial resolution of about 22 km and is driven by the global NCEP reanalysis (Kalnay et al., 1996) at its lateral boundaries. In addition, a spectral

nudging technique (von Storch et al., 2000) was applied. Here, differently from the conventional approach, the forcing was applied not only at the lateral boundaries but also in the interior of the model domain. This interior forcing was maintained by adding nudging terms in the spectral domain, with maximum efficiency for large scales and no effect for small scales (von Storch et al., 2000). From the hindcast hourly zonal and meridional wind components as well as sea level pressure were available and used to drive the hydrodynamic hindcast.

Additionally, and to maintain consistency with the original hindcast, the hydrodynamic model was driven by astronomical tides derived from the FES2004 global ocean tide atlas (Lyard et al., 2006). Tides were added at the lateral boundaries of the largest grid to account for the effect of tides on water levels and tide-surge interactions. The climatological monthly mean river run-off for 33 rivers was used, 10 of them are the Baltic Sea tributaries with the largest mean discharge. Model output (water levels) were stored hourly. To compare simulated water levels with observations, hindcast data from the grid points closest to

the selected gauge locations were used (red dots in Fig. 1). In addition, for the comparison of wind speed and direction, grid points off the coast are used (red asterisk in Fig. 1).





## 2.2 Observational data

To evaluate the results of the numerical simulations for the three surge events in the western Baltic Sea, hourly water level data from the gauges Travemünde, Warnemünde and Flensburg (Germany) (orange circles in Fig. 1) provided by Waterways and Shipping Administration, Germany (WSV, 2023) were used. In addition water levels from the gauge Landsort (Sweden) provided by the Swedish Meteorological and Hydrological Institute (SMHI, 2024) and from the gauge Degerby(Finland) provided by the Finish Meteorological Institut (FMI, 2024) were used. Water levels from Landsort and Dererby are frequently used as a proxy for the total water volume of the Baltic Sea and to estimate prefilling (e.g. Janssen et al. (2001), Hupfer et al. (2003), Mudersbach and Jensen (2009)). To evaluate the quality of the wind fields driving the hydrodynamic hindcast, observational data from the German Weather Service (DWD) for Boltenhagen near Travemünde, Warnemünde and Schönhagen - open coast location in the vicinity of Flensburg (orange asterisk in Fig. 1) were used (DWD, 2024). As hindcast data were available hourly, the data at full hours were used for comparison for both water level and wind, even where the observational data at higher frequencies were available.

## 3 Results

### 3.1 Description of the three storm surge events

The three severe storm surge events in 2017, 2019, and 2023 affecting the German Baltic coast showed very high peak water levels, while the wind speed during the events varied from moderate (2017, 2019) to strong (2023). The wind directions also differed between the events. While the events in 2017 and 2019 were dominated by northerly winds, easterly winds were prevailing during the most recent event in 2023.

In 2017, before the onset of the event, a high pressure system was located between the British Isles and Iceland (Lefebvre and Haeseler, 2017). As a consequence, the low pressure system responsible for the storm surge event in the western Baltic Sea took a relatively northern track from the Norwegian Sea over Scandinavia and the Baltic Sea toward northeastern Europe. Over the Baltic Sea associated wind fields initially came from westerly directions and changed towards more northerly and northeasterly directions over the southwestern Baltic Sea during the passage of the cyclone. During the event, the maximum wind speeds over the southwestern Baltic Sea were very moderate. They reached about 15 m/s 6-12 hours prior to the observed peak water levels, and only about 10 m/s at the times of surge maxima (Figure 2). However, observed surge levels were high (with a maximum of 1.73 m at Travemünde, 1.58 m at Warnemünde, and 1.76 m at Flensburg;) and led to the event being a once in 10-20 year event (Liu et al. (2022); https://stormsurge-monitor.eu). In the hindcast, both modeled wind and water level fields agree well with observations. Here, maximum surge heights are 1.76 m at Travemünde, 1.66 m at Warnemünde, and 1.76 m at Flensburg (Figure 2).

During the 2019 event, the large-scale atmospheric circulation pattern was comparable to that for the event in 2017. Again, an atmospheric high occurred over the British Isles and the low pressure system responsible for the storm surge event in the western Baltic Sea traveled along a track from northern Scandinavia, over Finland and towards the eastern Baltic States. The







**Figure 2.** Time series of water level (solid line), wind speed (dashed line) and wind direction (arrows) for observations (orange) and simulations (red) at Travemünde (top row) and Warnemünde (middle row) and Flensburg (bottom row) for the storm surge in 2017 (left column), in 2019 (middle column) and in 2023 (right column).



winds again initially came from westerly directions and subsequently turned toward northerly directions at the times of the maximum water level in the southwestern Baltic Sea. Peak surge heights were comparable to those during the 2017 event and reached values of 1.72 m in Travemünde, 1.65 m in Warnemünde, and 1.57 m in Flensburg (Figure 2). While wind directions were comparable to the 2017 event, wind speeds were higher (Figure 2). Again, hindcast wind and water levels agree well with observations with modelled peak surge levels reaching 1.75 m in Travemünde, 1.66 m in Warnemünde, and 1.57 m in Flensburg.

The most recent event on 20 October 2023 showed different characteristics. Here the large scale atmospheric circulation was characterized by a high-over-low weather pattern; that is, a blocking high pressure system over Scandinavia and a relatively stable low pressure system between the British Isles and west of France (DWD, 2023). In contrast to the other two events, this atmospheric situation lead to strong pressure gradients over the western Baltic Sea and correspondingly to strong easterly winds that have the potential to pile up water masses in the western Baltic Sea (Feistel et al., 2008). While the observed maximum water levels at Travemünde (1.75 m) and Warnemünde (1.46 m) were similar or even lower than during the previous two events, in Flensburg, a very high value of 2.25 m was reached, substantially exceeding the maxima of the 2017 and 2019 events. The hindcast peak surge levels of 1.68 m in Travemünde, 1.59 m in Warnemünde, and 2.19 m in Flensburg agree well with observations (Figure 2). Wind speeds during the event were substantially higher than in 2017 and 2019 with the simulated wind speed exceeding 20 m/s at all three locations. Hindcast and observed wind speed agree well in Travemünde and Flensburg, while in Warnemünde observed the wind speed is considerably lower than in the hindcast. Note that this offset in the wind speed comparison can be mostly explained by the fact that model grid points off the coast were used for comparison with observations at the coast as the only available in the area. The wind speeds off the coast are typically higher than over land. In addition, in 2023, as the dominant wind direction is from the east, the observational site is shadowed by the coast, which strongly enhanced the difference between land and open sea winds.

During the 2023 event, wind conditions were favourable for high water levels in the western Baltic Sea, with high atmospheric pressure over Scandinavia and low pressure south of the Baltic Sea. This weather pattern leads to strong atmospheric pressure gradients over the Baltic Sea that result in high wind speeds from the east. The situation was comparable to that during the devastating 1872 flood event in the western Baltic Sea (Bork et al., 2022). In contrast, the atmospheric situation during the 2017 and 2019 events was considerably different with prevailing moderate wind speeds from the North. In this case the fetch of the area of the wind influence was limited due to complicated topography of the Danish Straits and it suggests that during these events factors other than the prevailing wind field contributed substantially to the observed peak water levels. To better understand the role of different processes and their contributions to the observed high water levels, the water level components associated with these processes were estimated and analyzed separately and in the context of the total water level.



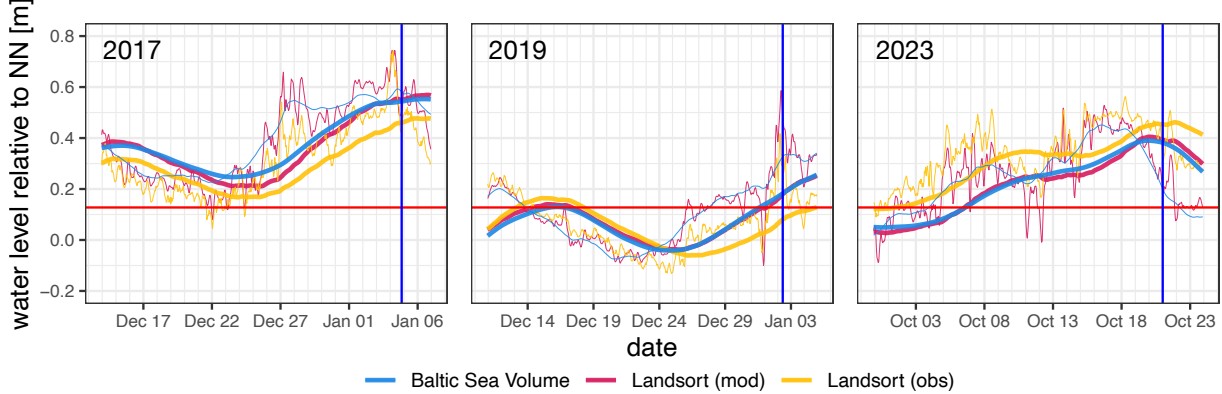

**Figure 3.** Hourly (thin lines) and 7-day average (thick lines) time series of Baltic Sea volume (blue), simulated water level at Landsort (red) and observed water level at Landsort (orange) before and around the time of the surge maximum (vertical blue lines) for event in 2017 (left), in 2019 (middle) and in 2023 (right). The horizontal red line represents the long-term mean volume of the Baltic Sea.

## 3.2 Main drivers contributing to the three storm surge events

### 3.2.1 Prefilling of the Baltic Sea

Inflow and outflow processes across the Danish Straits with characteristic timescales of about half a month or longer change the volume of the water in the Baltic Sea (Weisse et al., 2021). Such volume changes are often referred to as prefilling or preconditioning and lead to an increase or decrease of water levels in the Baltic Sea (e.g. Mudersbach and Jensen, 2009; Lehmann and Post, 2015). As such volume changes can not directly be inferred from observations, proxies are normally used. As a typical proxy for the prefilling in the Baltic Sea, water levels from the gauge Landsort (Sweden) are frequently used (e.g. Mudersbach and Jensen, 2009; Lehmann and Post, 2015).

In the present study a model simulation is available for the entire Baltic Sea basin, so that the prefilling or the Baltic Sea volume anomaly (BSVA) can be estimated directly. It is derived by summing the water volume anomaly of each grid cell, which is defined as the product of the area represented by each grid cell and the corresponding water level anomaly. It is finally divided by the total area of the Baltic Sea to obtain the sea level anomaly caused by prefilling. The estimated BSVA closely follows the modeled water levels at the grid cell closest to the Landsort location, apart from some high-frequency fluctuations (Figure 3).

Both time series are also comparable with the observations from the Landsort gauge (Figure 3), although there are some discrepancies. In particular, for the 2023 event the observed water level at Landsort is about 0.15 m higher than that simulated at the time of the storm surge maximum. When using the time series of the gauge station at Degerby (Finland), which has also been used as a proxy for the prefilling (e.g. Janssen et al., 2001; Bellinghausen et al., 2024), the water level is again relatively close to the simulated BSVA (Figure A1). Since we can determine the BSVA and thus the prefilling effect directly from the




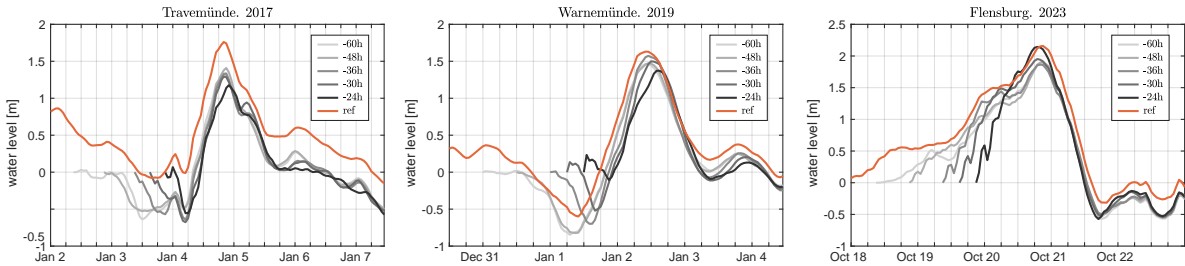

**Figure 4.** Water levels generated by short-term winds (starting 24h to 60h prior to the storm maxima, grey lines) without consideration of previous conditions. Water level time series for the three storms at the locations of the storm's maximum impact from reference simulation (red).

model data, and since we are interested in a consistent model-based analysis, we will use the BSVA instead of the Landsort water levels in the following.

In order to quantify the contribution of the prefilling to the total water level during the storms, the BSVA was additionally averaged over seven days (thick lines in Figure 3), so that the index represents the deviation from the mean state during one week prior to the water level event maximum. This appears to be a more relevant index than an instantaneous BSVA, as the latter is artificially uniform over the entire basin, whereas in reality the access of water is redistributed unevenly across the basin. This can be seen on the example of the storm of 2023, where the instantaneous BSVA is about 0.18 m lower than the

7-day average. In this case there is an outflow of the water through the Danish Straites, starting already several days before the storm maximum, which diminishes overall instantaneous BSVA. At the same time the rest of the prefilling is accumulated in the western Baltic Sea due to e.g. easterly winds but also the water swinging back in a seiche-like oscillations. Finally, the long-term mean Baltic Sea volume anomaly (0.13 m) was subtracted from the 7-day average to derive the BSVA contribution to the water level maximum. Thus, during the 2017 event, prefilling contributed about 0.41 m, which constitutes about 24%

of the peak water level in Travemünde, 25% in Warnemünde and 23% in Flensburg. During the 2019 event, the contribution from prefilling was negligible at about 0.05 m and accounting for only about 3% of the total peak water levels at all locations. For the 2023 event, the prefilling was about 0.25 m above average, thus contributing about 12% of the peak water level in Flensburg, 15% in Travemünde and 16% in Warnemünde.

### 3.2.2    Wind

Strong onshore winds pushing the water towards the coasts are usually the main driver of storm surges (Pugh and Woodworth, 2014). To disentangle the contribution of the local wind fields to the peak surge levels at the the three events, a set of model sensitivity experiments was carried out. For the three events, the TRIM-NP model was restarted from average (climatological) conditions 60 h, 48 h, 36 h, 30 h and 24 h prior to the timing of the peak surge levels in each event. In all experiments, the original atmospheric forcing from the multidecadal hindcast was used. This allowed an estimate of the contribution of the local



wind fields to the observed peak water levels could be obtained. The results are summarized in Figure 4 and each event is represented by one location for which the original storm surge was most prominent. For the event in 2017 the maximum surge levels caused by the short-term winds of different durations reached 1.41 m when the area was exposed to the wind influence for 60 hours (Fig. 4). This is by 0.35 m lower than the maximum water level obtained from the full model simulation. The underestimation holds both before and after the storm and corresponds well with the magnitude of the estimated Baltic Sea

prefilling observed and simulated for this event (Fig. 3). In this case the wind influence could explain about 80% of the water level elevation during the storm while the other 20% are attributable to prefilling and non-linear interactions.

The event in January 2019 showed a similar behavior, but the difference between the maximum water level from the full simulation and the short-term wind simulations were between 0.07 m and 0.16 m. This also is in accordance with the small prefilling rates of several centimeters (Fig. 3) for this event. Summarizing, the peak water levels in 2017 and 2019 were similar

(Fig. 2), the prefilling was lower in 2019 and the influence of the wind was slightly higher, consistent with slightly higher wind speeds in 2019. The combination of both effects explains the total water levels well. This shows that different combinations of the two main forcing factors can lead to similar extreme water levels. Moreover, even very similar atmospheric conditions may lead to different relative weighting of the contributions of these factors and to different resulting water levels.

The event in 2023 occurred under different atmospheric conditions, namely long lasting easterly winds, which alone could

cause up to 1.9 m surge levels at Flensburg with wind influence starting 60 hours prior to the water level maximum. Unlike previous examples, the model results for 2023 show higher water levels by shorter wind influence (dark grey vs. light grey lines in Figure 4). This can be partly explained by the fact that the same wind influence causes higher surge when the affected area has a smaller total water depths. In this case from "-60 h" simulation water elevation reached 1.3 m on 19 October at 21:00 when the "-24 h" simulation has started with zero water elevation, thus contributing 5-10% to the total water depth in the

area at this moment. Consequently, the subsequent water elevation in "-60 h" simulation was lower than in "-24 h" simulation when both were exposed to the same wind influence. Given that the positive water elevation 60 hours prior to the water level maximum is more realistic for this event, it is argued that the maximum wind-related contribution can be better represented by "-60 h" simulation. It demonstrates that there is an underestimation of maximum water level by 0.25 m when only wind conditions are considered compared to the full simulation. This is in fairly good agreement with the prefilling calculated for

the event, leaving the direct wind influence responsible for 85-90 % of the total maximum water levels.

### 3.2.3    Other processes

In addition to the effects mentioned above, water levels in the Baltic Sea are influenced by the atmospheric pressure and can experience variations in the decimeter range due to the inverse barometer effect (e.g. Leppärante and Myrberg (2009)). However, this process is related to the immediate atmospheric conditions and is directly considered by the hydrodynamic

model. Thus, in the remainder of this study the water levels referred to as wind-related do technically include the inverse barometer effect.

The Baltic Sea is a micro-tidal environment and tides can contribute 0.05–0.2 m to the water level elevation in the south-western Baltic Sea, depending on the location (e.g. Medvedev et al. (2016)). A tide-only simulation was used to estimate the





tidal signal during the event maximum. According to the simulations, the tidal component during the peak water level reached about 0.03 m in 2017 for Travemünde, -0.07 m in 2019 for Warnemünde and 0.1 m in 2023 for Flensburg. Although it is relevant to include tidal signal in the simulations to account for both the tidal elevation and tide-surge interactions, especially when single events are reconstructed and compared with observations, the tides can be neglected in the long-term analyses of extremes (e.g. Gräwe and Burchard (2012)). Therefore, in the present study, tides are not analysed explicitly, but are kept in the residual part of the water levels.

Another process that can contribute to the total water level variations is related to the seiches. These are standing waves that oscillate freely in the semi-enclosed basins under the influence of atmospheric conditions and persist even after the winds have ceased. In the Baltic Sea and its bays various modes of seiches with different periods have been observed or estimated (e.g.Leppäsrante and Myrberg (2009)). Although the phenomenon has been known and investigated for a long time, there are still controversial aspects about the main periods of the oscillation and whether they are basin-wide or rather a combination of local bay oscillations (see e.g. Jönsson et al. (2008) or discussion in Weisse et al. (2021)). This makes the clear separation of the seiche component rather uncertain, especially in the long-term perspective. This process is taken into account in the model simulations, but is not investigated here separately from a climatological point of view.

### 3.3    The recent storm surge events from a climate perspective

The discussed three events had surges ranked among the highest observed at the German Baltic Sea coast since the early 1950s. Furthermore, the events themselves occurred in close proximity over the past seven years.This raises questions about the extent to which these events were unusual, whether and how the frequency or severity of such events has changed over time, and how contributing factors such as prefilling or prevailing wind conditions may have changed. In the following, these questions are addressed using insights from the multidecadal hindcast.

### 3.3.1    How unusual were the events?

To place individual events in a historical context, their return periods are estimated. Several methods can be used to generate return value curves. In this case, the Generalised Extreme Value (GEV) method (Coles, 2001) based on annual maxima was applied to the 66 years of the hindcast data. The derived return values provide an indication of whether events fall within the expected range of extreme events or may be at the upper boundary of simulated and observed events (Figure 5). The estimated return values for the total water level show that the 2017 event at Travemünde is one of the highest in the simulation period with a return period of about 30 to 40 years. The 2019 event has a similar return period and the water level of 2023 still has a return period of 10 to 20 years. At Warnemünde, the 2019 event has a higher return period of more than 50 years, followed by the 2017 event with a similar return period, and the 2023 event still has a return period of almost 20 years. The analysis of the total water level at Flensburg shows that the 2023 event is an exceptional event in the simulation period with a return period of 66 years. Note, that the 66-year return period is the upper limit due to the length of the simulation period. Given the estimated return curve and the wider confidence interval, the water level would easily exceed the 100 year period. For Flensburg, the 2017 event is also one of the highest with a return period of almost 20 years, only the 2019 event, even if it falls





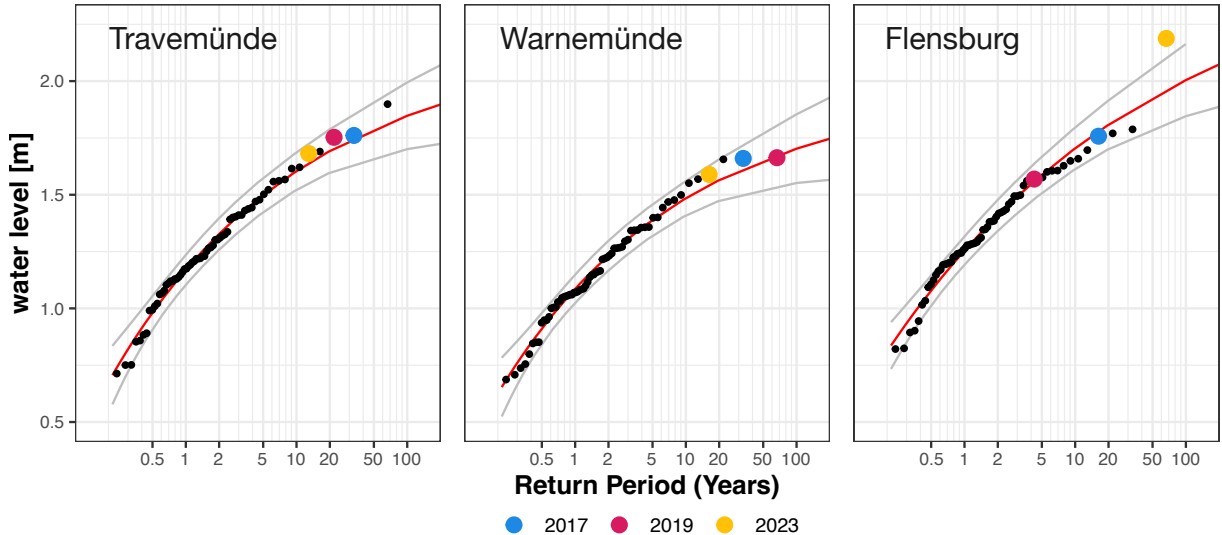

**Figure 5.** Return values (red curve) using GEV estimated from simulated annual maximum water levels (black dots) at Travemünde (left), Warnemünde (centre) and Flensburg (right). Grey lines indicate the 95% confidence interval of the return values. Maximum water levels of the recent events are represented by coloured dots (2017: blue; 2019: red; 2023: orange).

into the category severe storm surges, is a fairly normal event with a return period of about 4 to 5 years. It should be noted that the derived return value curves from the simulations differ from those estimated from observations (e.g. in Liu et al., 2022, https://stormsurge-monitor.eu). This may be due to the longer observation period than the modelling period. For example a

prominent in the southwestern Baltic Sea extreme event of 4 January 1954 is absent in the hindcast data. Another reason for discrepancies could be a slightly inaccurate representation of some event maxima by the simulation, or missing the absolute event maxima by the simulation if they occurred between full hours and are thus not included in the hindcast time series.

However, the question remains as to whether the contributing factors were also exceptional, and for which events which contributions were more crucial. In order to investigate this, the return value of the annual maximum of the BSVA (seven days

average) was analysed as an indicator of pre-filling (Figure 6). None of the BSVAs during the three events show a really high return period. Such, the BSVA for the 2017 event has a typical return period of one to two years. For the other two events (2019 and 2023), the BSVAs are still above the average BSVA. However, the return periods only range from weeks to months. This suggests that, under different circumstances and in combination with more extreme BSVA conditions, the total water levels could have been 0.3 m higher for the 2017 events and more than 0.5 m higher for the other two events.

When looking at the wind statistics, we are only interested in the strong winds responsible for the generation of extreme water levels, i.e. winds coming from a particular direction depending on the orientation of the coast. In order to calculate return values only for such wind components, the concept of the effective wind (Ganske et al., 2018) has been used. The effective wind is an orthogonal projection of the wind vector onto the prevailing surge-generating wind direction. Here, for simplicity, we used the wind direction that occurred during the highest water level event during the hindcast period at each of the three





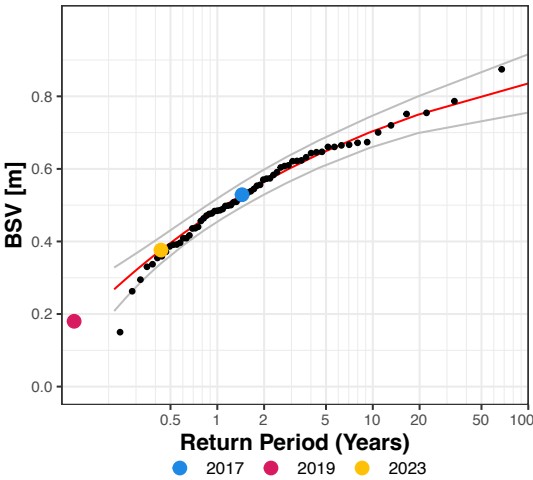

**Figure 6.** Return values (red curve) estimated with GEV from the simulated annual maxima of BSVA 7-days average (black dots). Grey lines indicate the 95% confidence interval of the return values. Average BSVA within 7 days prior to maximum total water levels of the recent events are represented by coloured dots (2017: blue; 2019: red; 2023: orange).

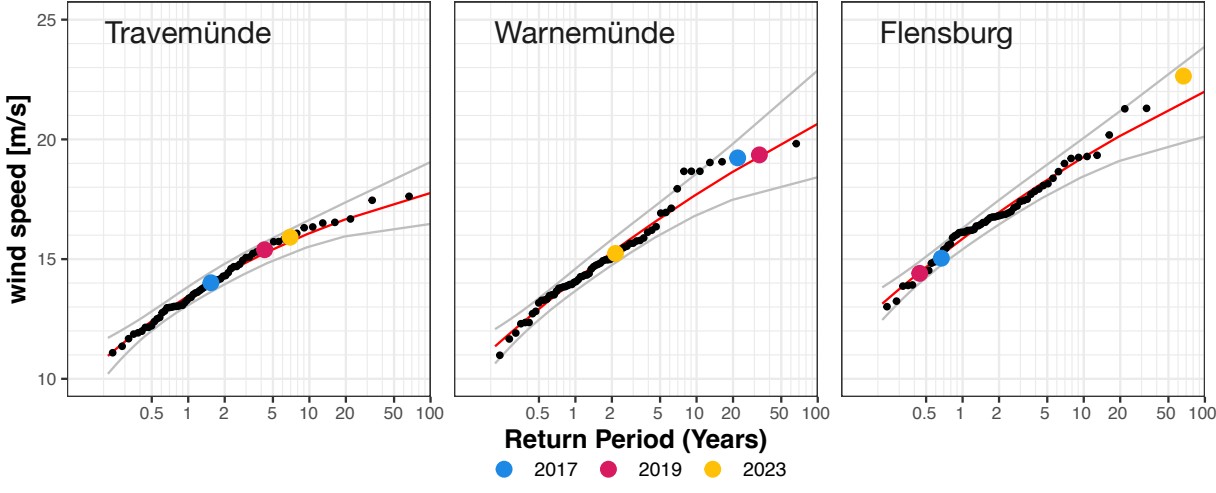

**Figure 7.** Return values (red curve) using GEV estimated from simulated annual maximum effective wind speed (black dots) for Travemünde at Lübecker Bight (left), for Warnemünde at Mecklenburger Bight (centre) and for Flensburg at Flensburger Fjord (right). The grey lines indicate the 95% confidence interval of the return values. Recent maximum effective wind speeds are represented by coloured dots (2017: blue; 2019: red; 2023: orange).





sites. Using this approach, the effective wind direction is northeast (42.6 °) at Travemünde, north (354.6 °) at Warnemünde and east (90.6 °) at Flensburg. With these wind directions the effective wind is derived and used to estimate the return values (Figure 7). This can then be used to analyse the extent to which the wind situation was exceptional for the three events in 2017, 2019, and 2023. For Travemünde the effective wind associated with the 2017 event can be expected on average once every 1-2 years. The return periods for the effective wind during the two events in 2019 and 2023 were still moderate, about 4 and 7 years

respectively. For Warnemünde the two events of 2017 and 2019 were highly unusual in terms of effective winds. Here such events could be expected once in 20 to 30 years. For Flensburg, the event in 2023 is outstanding also in terms of the effective wind. Here a return period of 66 years was estimated, which shows that the 2023 event in Flensburg was exceptional with the wind being the major contributing factor.

While for the 2019 event, wind speed and direction seem to be the main reason for the extreme high water levels, for the
other events, neither BSVA nor wind speed alone are reasonable explanations for such extreme water levels. A further analysis of the joint distribution of the event maximum water level and its two main contributors, BSVA and wind speed, could indicate the special nature of these events. Further analysis of three different periods will reveal whether the contribution of each factor has changed over time. Only water level events above 1 m are used for the joint distribution.

The joint distribution of the maximum water levels during each extreme event above 1 m and the BSVA averaged over 7
days prior to the event maximum is shown as a scatter plot for different locations (Figure 8). For Travemünde there are events with a higher BSVA than during the three events 2017, 2019, and 2023. None of these events has a total sea level exceeding the maxima of the three most recent events. There is, however, one event with a BSVA below average, that has a higher total water level. The event happened on 13.01.1987 and was characterized by strong winds (up to 25 m/s) from the north-east, which corresponds to the prevailing wind sector for generating maximum surge in Travemünde. This example indicates both,
the sensitivity of the surge height to the exact local wind directions and also a possibility for much higher surge magnitude in case such wind conditions coincide with the larger BSVA. Warnemünde shows a similar distribution of water levels and BSVA during extreme events. Notably, the 1987 event resulted in much lower water level here than in Travemünde, as the wind direction was not optimal for generating extreme surge in Warnemünde. At Flensburg, the 2023 event again shows its peculiarity with an extreme water level which has never occurred before and moderate BSVA. The 2017 event is characterised
by the relatively high BSVA and the 2019 event is not remarkable compared to other extreme events. When comparing different periods, there is no clear tendency towards more or less BSVA contribution to the extreme water levels. There is almost a slight decrease in the contribution of BSVA in the most recent period compared to the previous 20 years, with the eight highest BSVAs occurred during 1980-2001 period.

As the surge is generated over a period of time, it is not only the wind conditions at the time of the surge that are important,
but also the winds for at least several hours prior to the surge. Thus, the joint distribution of the total water level maximum and the wind speed and direction averaged over the six hours prior to the water level maximum was analysed (Figure 9). The wind speed during the 2017 event, was rather on the lower side of the distribution compared to all other extreme water level events (above 1.5 m). Compared to all severe storm surge events above 1.5 m, the 2017 event had the lowest wind speed at Travemünde and Warnemünde, but with rather optimal mean wind direction heading from north northeast towards the coast.



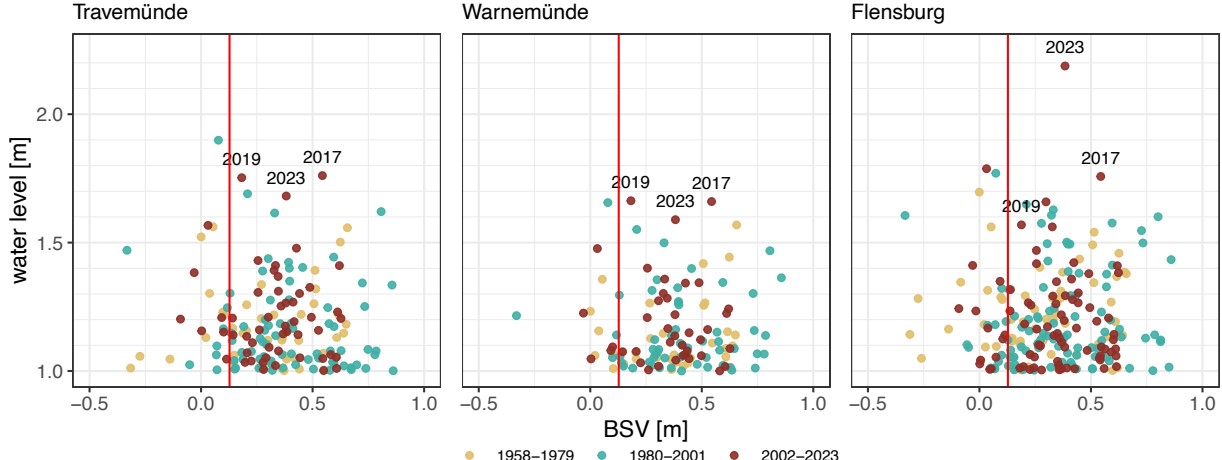

**Figure 8.** Scatterplot between water level and BSVA averaged over seven days prior to the event maximum at Travemünde (left), Warnemünde (centre) and Flensburg (right). Coloured dots indicate events from three 22–year long periods, the events discussed are annotated with their years. The red vertical line indicates the long-term mean BSVA.

Even at Flensburg, where there are some severe storm surge events with lower wind speeds, the wind speed on January 2017 was low compared to events with similar water levels. During the 2019 event, the wind speed at Travemünde was one of the highest during an extreme water level event, still high for Warnemünde and Flensburg, but not exceptional compared to other events. Both events show a wind direction with a northerly component, which tends to lead to high water levels at Travemünde and Warnemünde. The six hour mean wind before the 2023 event for Travemünde and Warnemünde was high, but comparable to other events with high water level. There was one event (13.03.1969) with the 6-hour-averaged wind speed and direction

comparable to those of 2023 event for Flensburg. However, the resulting water levels were significantly lower, partly because the BSVA was slightly negative in 1969, and partly because of lower peak wind speeds immediately before the event maxima.

There is no clear trend in the temporal variability of the joint occurrence of wind conditions and water level. While Travemünde shows a slight increase in the number of events with wind speeds above 15 m/s, these higher wind speeds do not

necessarily lead to an increase in the number of water level events above 1.5 m. At Warnemünde, events with even higher wind speeds than during the recent three events occurred in earlier periods. At Flensburg, with the exception of the 2023 event, wind speeds above 17 m/s occurred only in periods before the recent one.

### 3.3.2 Long-term changes in storm surge climate

To better investigate possible temporal variability and long term trends of extreme water level events, time series of hourly

hindcast water levels from 1958-2023 were analysed (Figure 10). The time series show pronounced short term variability at all three locations that may mask potential long-term changes. For the analysis of the variability of extremes, the storm surge classification used by the German Federal Maritime and Hydrographic Agency (Bundesamt für Seeschifffahrt und Hydrography, BSH) was applied. The BSH uses the following thresholds to classify storm surges in the Baltic Sea:



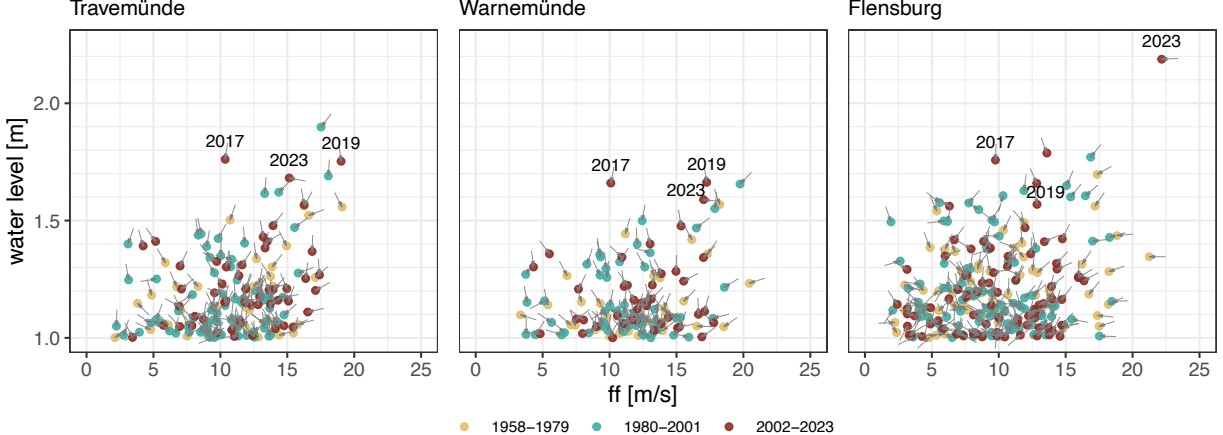

**Figure 9.** Scatterplot between water level maximum and wind speed averaged over the six hours before the event maximum at Travemünde (left), Warnemünde (centre) and Flensburg (right). Coloured dots indicate three different 22-year periods, the events discussed are annotated with their years. Arrows indicate the average wind directions six hours before each event.

1. storm surge: an event with a peak water level between 1 m and 1.25 m above the mean water level (MW),

2. medium storm surge: an event with a peak water level between 1.25 m and 1.5 m above MW

3. severe storm surge: an event with a peak water level between 1.5 m and 2 m above MW

4. very severe storm surge: an event whose peak water level exceeds 2 m above MW

As only the 2023 event exceeds the threshold for a very severe storm surge, only the other three categories are used in the analysis and the 2023 event is treated as a severe storm surge.

The storm surge classification was superimposed onto the hourly water level time series (Figure 10). The sequences of storm surges in the different intensity classes show periods with lower and higher values. For example, lower values occurred in the late 1960s and early 1970s, while surges became more frequent in the late 1980s to mid-1990s.

The time series further reveals that the three events 2017, 2019, and 2023 were among the highest in the simulation period. For Flensburg the event in 2023 was the largest during the hindcast period starting in 1958. This cluster of three events in the last seven years indicates an increase in the number of events above 1.5 m above MW, but this positive trend is subject to great uncertainty. Furthermore, for the two lower categories, storm surge (1) and medium storm surge (2), there is no long-term trend for all three locations. Similar analysis of the time series of the BSVA and the wind speed shows high short-term variability and small linear trends that are accompanied by large conference intervals (see Figure B1 and Figure B2 in the appendix).





**Figure 10.** Simulated time series of hourly water levels for the period 1958 to 2023 at Travemünde (top), Warnemünde (middle) and Flensburg (bottom). Storm surge events between 1 m and 1.25 m are marked in blue, medium surge events between 1.25 m and 1.5 m in red and severe surge events exceeding 1.5 m in orange. Colored lines represent linear trends calculated for the respective type of events with the 95% confidence interval.



## 4 Conclusions

Three recent extreme storm surge events in the (south)western Baltic Sea were analyzed and put into a climate perspective. The comparison of water levels at three locations along the German Baltic Sea coast showed generally good agreement between the hindcast model simulations and observations. Both datasets confirm that the three events were among the highest at the German Baltic Sea coast at least within the period from 1958-2023. We further focused on the relative contributions of the two key factors associated with storm surges in the Baltic Sea, namely the prefilling caused by the inflow of water masses from the

North Sea during days to weeks prior to the event and a component induced by the short-term wind influence during the storm.

The events in 2017 and 2019 can be associated with a high pressure located over western Europe (British Isles, France), which led to a storm track across Scandinavia and towards eastern parts of Europe, resulting in northerly winds in the southwestern part of the Baltic Sea. During the 2017 event, the wind speed was relatively moderate, but the Baltic Sea Volume was relatively high, constituting about 25 % of the total maximum water level. The 2019 event showed high wind speed, and, espe-

cially in Warnemünde, the wind direction (from the north) was rather favourable for creating high water levels at the coast. The contribution of prefilling was small, in the order of 3%, during the 2019 event. The 2023 event was dominated by extremely high wind speeds caused by a Scandinavian blocking situation. The high pressure over Scandinavia and the low pressure system between France and the British Isles lead to prevailing easterly winds over the western Baltic Sea, resulting in the highest water levels in the simulation period at one location (Flensburg). This is also confirmed by observations, and in the case of the

2023 event, the water level was the highest observed in the last 100 years (Kiesel et al., 2024). The prefilling during this event was moderate but still it contributed 10-15 % of the total maximum water level depending on the location.

Additionally, both contributing factors were analysed in the context of historical storm events based on the model simulations for the past six decades. It can be stated that for the 2017 and 2019 events, the wind speed was not the highest during the storm event, especially when compared to other high water level events. Again, the 2017 event is relatively unique as the wind speed

shortly before and during the event was only about 10 m/s and thus lower than for all other events with surge heights above 1.5 m within a 66-year period, where wind speeds near Travemünde varied between 12.5 m/s and 18 m/s. In contrast, the event of 2023 showed extreme water levels and extreme wind speeds unprecedented within at least the considered 66 years.

The analysis of the effective wind based on the wind directions favourable for the generation of high water levels at specific locations, showed that the return periods for the 2017 and 2019 events at Warnemünde and the 2023 event at Flensburg were

high - from 20 to at least 66 years. The same storm events caused much smaller effective winds for other locations with the return values corresponding to 1-7 years return period. This is reflected in the water level magnitudes at different parts of the coast and underlines the importance of accurate wind direction representation, which is particularly important for predicting extreme water levels. It can be further concluded that while the total water levels of the recent events are among the highest in past decades, with a return period of 10 to 66 years, the Baltic Sea Volume magnitudes were rather normal, with the return

period of one and half year or shorter. It should be noted that calculating and estimating return values based on a limited sample size, such as 66 years, has limitations. This is particularly true for extreme events at the edge of the sample size, as the results may be uncertain and have a wide confidence interval. This suggests that longer return periods are possible, especially



when compared to return periods estimated from observations (e.g.Liu et al. (2022); https://stormsurge-monitor.eu, Kiesel et al. (2024)).

The multidecadal analysis of the water level maxima exceeding 1m showed that there is no clear trend in the evolution of extremes within the past six decades and that the interannual variability of extreme water level events in the southwestern Baltic Sea is rather high. This is in line with the findings of Lorenz and Gräwe (2023) and Feser et al. (2015), who discuss the variability of extremes and unclear trends when considering different historical periods. The same holds for the contributing components of the extreme water levels, where we found no increase in the frequency or magnitude of BSVA in the past

decades but rather a small decrease in the last 20 years. No significant changes were observed for wind either (Bierstedt et al., 2015). Considering the non-exceptional nature of the contributing components during the recent events (with the exception of the wind conditions for Flensburg in 2023), it is probable that a combination of higher individual contributions can co-occur, potentially leading to higher total water levels under the present-day conditions, in line with the previous conclusions by e.g. Andrée et al. (2023).

Finally, it should be noted that the exception of these events was the combination and simultaneous occurrence of their individual contributions. Especially since some of these events could have had a higher water level with a different co-occurrence of high BSVA and high wind speed with favourable wind direction, this shows the importance of analysing compound events.





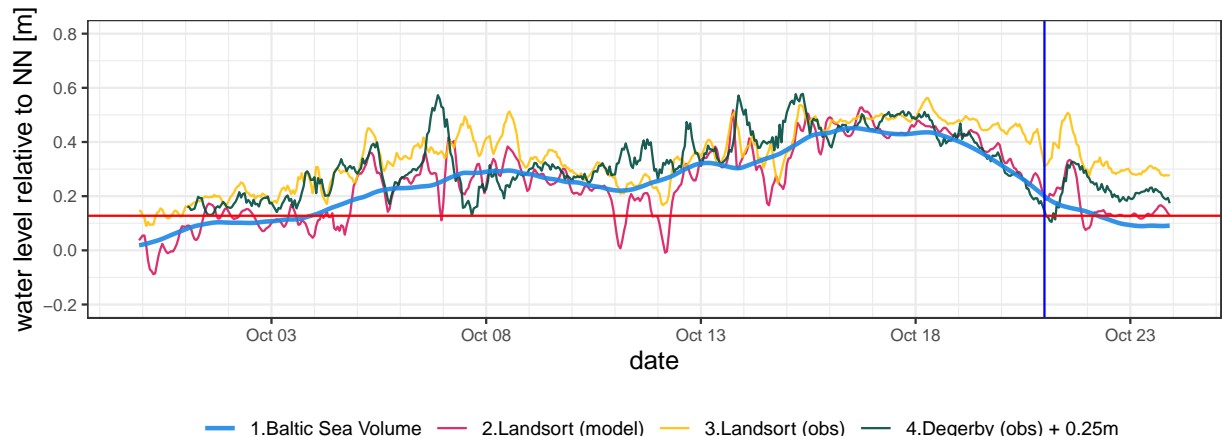

**Figure A1.** Time series of Baltic Sea volume (blue), simulated water level at Landsort (red), observed water level at Landsort (orange) and observed water level at Degerby (green) before and around the time of the surge maximum (vertical blue lines) for event in 2023. The horizontal red line represents the long-term mean volume of the Baltic Sea.

## Appendix A: Prefilling of the Baltic Sea

## Appendix B: Long-term changes in storm surge climate

*Author contributions.* NG was responsible for the analysis and writing the text. LG was responsible for running the model simulation and writing the text, RW was responsible for the general idea of the analysis and writing the text.

*Competing interests.* The authors declare no competing interests.

*Acknowledgements.* The authors thank E.M.I. Meyer and I. Grabemann for their discussion on during the research and writing process for this study.







**Figure B1.** Simulated time series (grey) of the hourly Baltic Sea Volume Anomaly (BSVA) for the period 1958 to 2023 at Travemünde (top), Warnemünde (middle) and Flensburg (bottom). BSVA during storm surge events between 1m and 1.25m are marked in blue, medium surge events between 1.25m and 1.5m in red and severe surge events exceeding 1.5m in orange. The grey line represents the linear trend for hourly BSVA, the coloured lines represent the linear trends calculated for each type of event with the 95% confidence interval.





**Figure B2.** Simulated time series (grey) of hourly wind speeds for the period 1958 to 2023 at the offshore site for Travemünde (top), Warnemünde (middle) and Flensburg (bottom). Wind speed during storm surge events between 1m and 1.25m are marked in blue, medium surge events between 1.25m and 1.5m in red and severe surge events exceeding 1.5m in orange.The grey line represents the linear trend for hourly wind speed, and the coloured lines represent the linear trends calculated for each type of event with the 95% confidence interval.



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
