# Peer review of "Recent Baltic Sea Storm Surge Events From A Climate Perspective"

_EGUsphere, 2024_

## Author Response (AR1)

Authors Response to the comments of the anonymous reviewer **RC1** on the manuscript *egusphere-2024-2664*

**„Recent Baltic Sea Storm Surge Events From A Climate Perspective"**

by Nikolaus Groll, Lidia Gaslikova, and Ralf Weisse

We thank the anonymous reviewer#1 for the comments, which helped to improve the manuscript. In the following, the reviewers comments are shown in blue. The authors response will be under each comment and suggested changes in the text will be in *italic.*

This manuscript is a nice piece of analysis of three recent dangerous storm surges in the western Baltic Sea, an area famous for extensive variability of drivers of water level, their combinations and resulting properties of high water level events. The main outcome is that two of these events, even though seemingly severe, belong to the pool of relatively frequently occurring episodes in contemporary climate while the third one has at least one truly unusual feature in terms of wind direction. This outcome is in line with the general perception of the nature of climate change in the Baltic Sea region. Namely: storms have not become systematically stronger in this area. Instead, most severe events are driven by specific combinations of various drivers. Another important message is that wind direction may become the most critical feature in development of extreme events.

The manuscript is written professionally, with very good command of English. The setup of the problem is clear, the used methods are described properly and applied correctly, and statistical methods are employed adequately. The images are clear and informative. The conclusions are firmly backed up by the analysis. It is thus my pleasure to recommend this manuscript for publication, possibly with marginal technical revisions.

The reviewer has suggested minor corrections to the text. Most of them are either typos or clarifications, we agree with all of them and will correct them in the revised version.

The reviewer noted that there is a discrepancy between the estimated return values and the observations, which is also described in the literature and that the annual maxima may not be independent.
Following this suggestion, we also calculated returns based on July to June values and found some small differences. By using the block maxima from July to June to derive the return values, we lose the October 2023 event for the estimation of our return values, so we still use the annual (calendar) maxima for our calculation. However, we add some text and discuss this source of possible differences between the results presented and others in the revised version.

Line 4: remove "of".
„or of prefilling of the Baltic Sea" changed to *„or prefilling of the Baltic Sea"*

Line 5: remove either "hindcast" or "simulation".

changed to *„A numerical hindcast is used to"*

Line 8: it makes sense to add a couple of words to "water level" to explain what is meant.

combination of *„atmospheric induced"* water level *„changes"* and

Line 15-16, 42, 48-49 and in some other occasions below: please check the sequence of references.

cited references are put in chronological order
*L15: (Suursaar et al., 2006; Suursaar and Sooäär, 2007; Männikus et al., 2019)*

*L42: (e.g. Wübber and Krauss, 1979; Otsmann et al., 2001; Jönsson et al., 2008)*

*L48: (e.g. Suursaar et al., 2006; She and Nielsen, 2019; Aakjær and Buch, 2022; Kiesel et al., 2024)*

*L372: Feser et al. (2015)  and Lorenz and Gräwe (2023)*

Line 19: it is recommended to use MWL.

changed to *„MWL"* and in Line 319,320,321,322 and 330

Line 27 and in many occasions below: perhaps "south-western" is more traditional.

changed to "*south-western*" and in Line 17, 46, 60, 99, 100, 110, 250, 335 and 372

Line 28: remove "that" and "occurred" for brevity.

changed to *„winds persisted for two days and reached peak wind speeds of 102 km/h (Kiesel et al., 2024)."*

Line 37: I guess that the authors actually have in mind the publication [Soomere, T., Eelsalu, M., Kurkin, A., Rybin, A. 2015. Separation of the Baltic Sea water level into daily and multi-weekly components. Continental Shelf Research, 103, 23–32, doi: 10.1016/j.csr.2015.04.018]. The paper (Soomere and Pindsoo, 2016) made use of the 8-day time scale detected in the previous paper.

We add the reference Soomere et al. (2015) and add some clarification on length of prefilling events.

*„Typically, such variations that may lead to a prefilling (Lehmann and Post, 2015; Andrée et al., 2023) of the Baltic Sea have timescales of about 8 days (Soomere et al 2015 et al.) or even longer from week to even month in same cases (Soomere and Pindsoo, 2016) and …"*

Line 43: perhaps "unfavourably" would sound better.

changed to „*When unfavourably coupled with storm surges*…"

Line 60: consider replacing "last" by "finest" or similar.

changed to „*… and the finest nest covers* …"

Line 82: must be Degerby.

changed to „*Degerby*"

Caption to Figure 2: replace "and" by comma before ''Warnemünde''.

changed to „*…Travemünde (top row), Warnemünde*…"

Lines 130-132 mostly repeat information presented on lines 115-120.

We totally agree and removed the first parts of L130 to L132

Line 198: something is wrong with the end of the sentence.

We rearranged the sentence

„*Thus, on 19 October at 21:00 the water levels for "-60h" simulation reached 1.3m, whereas at the same time the "-24h" simulation started with zero water level. As a result, the subsequent wind affected 1.3m deeper water in case of "-60h" simulation than in case of "-24h" simulation, which could account for about 5-10\% reduction in surge height for the former.*"

Line 221: "persist even after the winds have ceased" is not entirely correct as in many occasions the seiche is only launched when the wind starts to decay.

change to
„*…and persist even after the winds have ceased or even just started to decay.*"

Line 247: "of severe storm surges"

changed to
"*… of severe storm surges*"

Lines 247-249: it seems that that return value curves and values of parameters of extreme value distributions estimated from measured and simulated water levels deviate systematically in many locations of the Baltic Sea. Eelsalu et al. (2014) [Eelsalu, M., Soomere, T., Pindsoo, K., Lagemaa, P. 2014. Ensemble approach for projections of return

periods of extreme water levels in Estonian waters, Continental Shelf Research, 91, 201–210, doi: 10.1016/j.csr.2014.09.012] hypothesize that wave set-up could be one of reasons while Soomere et al. (2018) [Soomere, T., Eelsalu, M., Pindsoo, K. 2018. Variations in parameters of extreme value distributions of water level along the eastern Baltic Sea coast. Estuarine, Coastal and Shelf Science, 215, 59–68, https://doi.org/10.1016/j.ecss.2018.10.010] demonstrate substantial mismatch of estimates of parameters of the generalized extreme value distribution retrieved from modelled and measured water level time series. Anyway, this is a minor and almost irrelevant aspect in the context of this manuscript.

We added one line to further point to this differences between observed and models estimates.

*L252: „This discrepancies between return value estimated from observations and from simulations is also found and discussed in detail by others (e.g Soomere et al. 2018)."*

Soomere, T., Eelsalu, M., Pindsoo, K. 2018. Variations in parameters of extreme value distributions of water level along the eastern Baltic Sea coast. Estuarine, Coastal and Shelf Science, 215, 59–68, https://doi.org/10.1016/j.ecss.2018.10.010]

Line 236: It might be mentioned that the some pairs of annual maximum water levels are not necessarily independent in the Baltic Sea because of possible prefilling covering December and January. This feature may slightly affect applicability of the generalised extreme value distribution in the interior of the Baltic Sea. For this reason some authors recommend using maxima over windy autumn and winter season that are definitely independent. However, the possible difference in the results apparently would be very minor.

As reviewer #2 also pointed to this fact and wanted to shift the description of the GEV, and a section with the methodology is introduced

*2.3.2 GEV*

*Several methods can be used to estimate return periods. Here, the Generalised Extreme Value (GEV) method (Coles, 2001) based on annual maxima was applied to the 66 years of hindcast data. In order to calculate the GEV distribution, block maxima have to be derived over a certain time period. The definition of the time period is depending on variable. For wind and wind-related variables, such as extreme water levels, the storm season between autumn and spring (Eelsalu et al. 2014) or summer and the following summer (Liu et al. 2022) is often used. Especially for the Baltic Sea, possible prefilling at the end of the year can also influence extreme water levels in the following year, which would then cause dependent variables. Here, however, we have chosen the calendar year (January to December) to derive the block maxima. One reason is that the extreme event of interest occurred in October 2023, and as our dataset ends in 2023, we would not have a full storm season 2023-2024 and be missing this event. A comparison with the results using block maxima from July to June shows only minor differences at the longer periods (not shown).*

Line 426: must be Küste.

*correct to „Küste"*

Line 456: must be Leppäranta.

corrected to „ *Leppäranta*"

Lines 474, 486, 488: must be Suursaar, Ü.

corrected to „*Suursaar,Ü*" accordingly

Line 494: must be Dailidienė.

corrected to „*Dailidienė*"

Authors Response to the comments of two anonymous reviewer **RC2** on the manuscript *egusphere-2024-2664*

**„Recent Baltic Sea Storm Surge Events From A Climate Perspective"**

by Nikolaus Groll, Lidia Gaslikova, and Ralf Weisse

We thank the anonymous reviewer#2 for the comments, which helped to improve the manuscript. In the following, the reviewers comments are shown in blue. The authors response will be under each comment and suggested changes in the text will be in *italic.*

The paper is well-written and addresses the important issue of extreme water levels in the western Baltic Sea. The analysis examines the various components contributing to extreme water levels, with a focus on the role of wind speed and direction in driving storm surges and prefilling of the Baltic Sea. The separation and detailed analysis of these components provide valuable insights into the severity of individual events and the combined effects of extreme water level components. The study demonstrates that while the water levels in 2017, 2019, and 2023 were among the highest on record, the individual components were far from their potential extreme values (with the exception of Flensburg in 2023). This indicates that these events, despite their historical significance, did not reflect the full potential severity of the contributing processes.

The paper also effectively highlights the critical role of storm direction in driving high storm surges. For instance, although wind speeds in 2019 were higher at Travemünde and Warnemünde, water levels in these locations did not exceed those observed in 2017, likely due to small but significant differences in wind direction.

While the paper comprehensively addresses the key drivers of extreme water levels, the omission of wave-induced set-up is notable. Wave set-up can contribute significantly to extreme water levels, adding approximately 20 to 25% of the significant wave height to the water level depending on the method of estimations. During wind speeds of 15–20 m/s, wave heights were likely substantial, especially when the effective wind direction was near perpendicular to the shoreline. Although detailed wave-induced set-up analysis for these events is not expected, discussing this phenomenon in the context local features would contribute the study. Additionally, providing some information on wave conditions during these events would be beneficial.

Several methodological and technical details, such as return period estimations and effective wind speed calculations, are described/mentioned under the results section. It is recommended to move these descriptions to the methods section (Section 2) to improve the structure and clarity of the paper.

The reviewer commented that while the study addresses several key factors that contribute to the total water level during extreme events, the study overlooks the potential contribution of wave set-up to the total water level. We agree with the reviewer that the effect of wave set-up is important and should not be overlooked in the discussion. In the revised version, the importance of wave set-up for the total water level is discussed, but a detailed analysis of wave set-up is beyond the scope of this study, especially as for a comprehensive calculation of wave set-up it is important to know the slope of the seabed in front of the particular coast and some higher-resolved spatial wave information  which is currently not available to the authors.

Specific remarks:

Line 15: References should be arranged in chronological order. Correct order in (Suursaar and Sooäär, 2007….etc).

as also suggested by RC1, this and other references are put in chronological order

*L15: (Suursaar et al., 2006; Suursaar and Sooäär, 2007; Männikus et al., 2019)*

*L42: (e.g. Wübber and Krauss, 1979; Otsmann et al., 2001; Jönsson et al., 2008)*

*L48: (e.g. Suursaar et al., 2006; She and Nielsen, 2019; Aakjær and Buch, 2022; Kiesel et al., 2024)*

*L372: Feser et al. (2015)  and Lorenz and Gräwe (2023)*

Line 20: Provide the water level range for all three storm events, as was done for the 2017 storm (e.g., peak water levels of 1.6 to 1.8 m). Additionally, add "Fig. 1" after the first location mentioned in the text.

Based on the reports from the BSH (Bundesamt für Seeschifffahrt und Hydrographie, the Federal Maritime and Hydrographic Agency in Germany) we added the range of the observed water level form tide gauges along the German Baltic Sea coast. According to these reports the maximum water level ranges from 1,39 m to 1,83 m for the 2017 event (BSH 2017), from 1,41 m to 1,91 m for the 2019 event (BSH 2019) and from 1,08 m to 2.27 m above MWL for the 2023 event (BSH 2023).
This will be add to the revised version of the manuscript, additionally Fig. 1 is mention earlier

*L19: „along the German Baltic Sea coast (Fig. 1)."*

BSH: Sturmflut vom 04./05.01.2017, https://www.bsh.de/DE/THEMEN/Wasserstand_und_Gezeiten/ Sturmfluten/_Anlagen/Downloads/Ostsee_Sturmflut_20170104.pdf?__blob=publicationFile&v=2 accessed: 5-2-2025, 2017.

BSH: Sturmflut vom 02.01.2019, https://www.bsh.de/DE/THEMEN/Wasserstand_und_Gezeiten/Sturmfluten/ _Anlagen/Downloads/Ostsee_Sturmflut_20190102.pdf?__blob=publicationFile&v=2#:~:text=Das BSH gab die erste,über dem mittleren Wasserstand gemessen. accessed: 5-2-2025, 2019.

BSH: Schwere Sturmflut vom 20.Oktober 2023, https://www.bsh.de/SharedDocs/Downloads/DE/Schwere-Sturmflut-20-Oktober-2023.pdf?__blob=publicationFile&v=2. accessed: 5-2-2025, 2023.

Line 35: Prefilling timescales of about 8 days was evaluated by T. Soomere et al. (2015) (https://doi.org/10.1016/j.csr.2015.04.018), while Soomere and Pindsoo (2016) demonstrated that the Baltic Sea's background water level might remain elevated for weeks or even months in some cases.

As also mentioned by RC1, we add the reference Soomere et al. (2015) and add some clarification on length of prefilling events.

*„Typically, such variations that may lead to a prefilling (Lehmann and Post, 2015; Andrée et al., 2023) of the Baltic Sea have timescales of about 8 days (Soomere et al 2015 et al.) or even longer from week to even month in same cases (Soomere and Pindsoo, 2016) and …"*

Line 75: Why modelled wind data points were chosen that far? Please provide some explanations. For example Flensburg, both observed and measured wind data points are quite far and modelled wind data point is located vicinity of one island, that may affect the wind fields in that data point? Please provide some explanations of the choice of the data points.

The locations selected for comparison from the model data are indeed located further from the coast. Considering the spatial resolution of the used regional atmospheric model being 22 km, this was a necessary compromise to ensure the wind data were relatively undisturbed by the complex land-sea interactions not fully resolved by the model.

In the revised version, there will be a better explanation of why the sites are chosen.

*Line 75: „In addition, for the comparison of wind speed and direction, grid points off the coast are used (red asterisk in Fig. 1). The locations far off the coast are chosen, because of the coarser land-sea mask of the forcing regional atmospheric model, where some small islands and a complex coastline are not resolved, which influences the land-sea interactions in the area. As the primary interest is on the wind impact over the open sea, this is considered to be a good compromise, given the observational data available near the coast."*

*Line 85: „…observational data (orange asterisk in Fig. 1) from the German Weather Service (DWD) for Boltenhagen near Travemünde, Warnemünde and Schönhagen were used (DWD, 2024). Note that location Schönhagen was used for Flensburg, as the wind observation at Flensburg has missing data around the event maxima and is more influenced by land-sea interaction, making it more difficult to compare with the location used in the simulation."*

Line 85: Specify if the "full hour" data represents hourly mean values.

A more precise definition of what the hourly data at every full hour represent will be given. The wind speed at every full hour represent the mean of the last model time step. This fits the observational values, which is typical the mean over the last ten minutes before the reported time step.

*Line 86: „As hindcast data were available hourly, the data at full hours were used for comparison for both water level and wind, even where the observational data at higher frequencies were available.The simulated wind speed at each full hour represents the instantaneous value of the last model time step, which is adjusted to be comparable to the*

*observational values, which are typically the mean of the last ten minutes before the full hour."*

Paragraph 3.2.1 (Prefilling): Consider to move the methodological description for prefilling extraction to Section 2. Briefly explain the choice of the 7-day average.

We shift the description to section 2 and explain the chosen length.

*2.3. Methods*

*2.3.1 Prefilling*
*Inflow and outflow processes across the Danish Straits with characteristic timescales of about half a month or longer change the volume of the water in the Baltic Sea (Weisse et al., 2021). Such volume changes are often referred to as prefilling or preconditioning and lead to an increase or decrease of water levels in the Baltic Sea (e.g. Mudersbach and Jensen, 2009; Lehmann and Post, 2015). As such volume changes can not directly be inferred from observations, proxies are normally used. As a typical proxy for the prefilling in the Baltic Sea, water levels from the gauge Landsort (Sweden) are frequently used (e.g. Mudersbach and Jensen, 2009; Lehmann and Post, 2015). Additionally the time series of the gauge station at Degerby (Finland) has also been used as a proxy for the prefilling (e.g. Janssen et al., 2001; Bellinghausen et al., 2024). In the present study a model simulation is available for the entire Baltic Sea basin, so that the prefilling or the Baltic Sea volume anomaly (BSVA) can be estimated directly. It is derived by summing the water volume anomaly of each grid cell, which is defined as the product of the area represented by each grid cell and the corresponding water level anomaly. It is finally divided by the total area of the Baltic Sea to obtain the sea level anomaly caused by prefilling. As the prefilling of the Baltic Sea took place over a longer period, days to weeks, a moving average over several days is usually used to describe the amount of the prefilling. Here we followed Soomere et al. 2015 with a multi-day average (8 days) and using a seven day moving average for describing the amount of prefilling in the Baltic Sea.*

Line 185: Were the contributions of 80% (wind) and 20% (prefilling) consistent across all locations?

For locations with comparable total water level maxima, the shares for wind and prefilling remain similar. For locations with lower total water levels (e.g. Warnemünde for 2017 event), the absolute magnitude of prefilling remains and thus the relative contribution of the wind decreases to 70%. This is also supported by the sensitivity experiments. Explanation will be added to the text.

Line 200: For the 2023 event, specify the wind's contribution to water levels at other locations for comparison.

During this event Flensburg experienced "perfect" wind condition for generation of maximum water levels. For the locations farther to the south the wind direction was not so optimal, this resulted in lower total water level maxima (from observations and the model).

The sensitivity experiments confirm lower relative contribution by the wind-induced water levels of about 80-83%. This will be added to the text.

Line 210–225:Among other contributions I would also expect a short discussion of the potential contribution of wave set-up, considering wave conditions (significant wave height and wave directions respect to the shoreline orientation) during these storms.

We will give a short description of the importance both the wave set-up, but no detailed analysis of the contribution of the wave set-up will be given as this is beyond the scope of this study.

*L227: „An additional process that contributes to the overall water level in coastal regions is the wave set-up. As the wave breaks in the surf zone, the momentum associated with the wave is transferred to the water column, causing an increase of water level towards the coast. Depending on the wind and wave situation during an extreme storm event and the coastal location, the wave set-up can contribute up to 30% of the total water level (Eelsalu et al. 2014) or up 35% at Australian coast (Hetzel et al. 2024). Su et al. (2024) estimated the wave-induced set-up along the Danish coast and found that for the 2017 event the contribution along the western Baltic Sea was rather small with only 5 cm, which corresponds to less than 5% of the total water level. For the 2019 and 2023 events, the contribution of wave set-up could be larger as the wind speed and wave height were higher compared to 2017. However, in order to estimate the contribution to the total water level, knowledge of the bed slope along the coastline and sufficiently resolved wave information is required. As both types of information are beyond the scope of the present study, the contribution of the wave set-up to the total water level is not addressed.“*

Eelsalu, M., Soomere, T., Pindsoo, K., Lagemaa, P. 2014. Ensemble approach for projections of return periods of extreme water levels in Estonian waters, Continental Shelf Research, 91, 201–210, doi: 10.1016/j.csr.2014.09.012

Hetzel, Y., Janeković, i., Pattiaratchi, C., Haigh, I. , 2024. The role of wave setup on extreme water levels around Australia. Ocean Engineering, 308, 118340, doi: 10.1016/j.oceaneng.2024.118340.

Su, J., Murawsk, J., Nielsen, J.W., Madsen K.S., 2024. Coinciding storm surge and wave setup: A regional assessment of sea level rise impact, Ocean Engineering, 305, 117885, doi:10.1016/j.oceaneng.2024.117885.

Line 235: I recommend to describe statistical methods (e.g., GEV assumptions, annual maxima) in Section 2. Are applied annual maxima values uncorrelated in GEV?

A short subsection describing the statistical methods used will be added to Section 2. A brief description will also be given of how and why the annual maxima are calculated, and the possible differences in choosing a different time period to define the block maxima for GEV.

*2.3.2 GEV*

*Several methods can be used to estimate return periods. Here, the Generalised Extreme Value (GEV) method (Coles, 2001) based on annual maxima was applied to the 66 years of hindcast data. In order to calculate the GEV distribution, block maxima have to be derived over a certain time period. The definition of the time period is depending on variable. For wind and wind-related variables, such as extreme water levels, the storm season between autumn and spring (Eelsalu et al. 2014) or summer and the following*

*summer (Liu et al. 2022) is often used. Especially for the Baltic Sea, possible prefilling at the end of the year can also influence extreme water levels in the following year, which would then cause dependent variables. Here, however, we have chosen the calendar year (January to December) to derive the block maxima. One reason is that the extreme event of interest occurred in October 2023, and as our dataset ends in 2023, we would not have a full storm season 2023-2024 and be missing this event. A comparison with the results using block maxima from July to June shows only minor differences at the longer periods (not shown).*

Line 265: Please provide more explanation of effective wind speed. I understand this is derived from the highest storm surge data and you have given the direction for each site that corresponds to the effective wind directions. What range was considered of extraction of effective wind from the given directions? Was it e.g. ±10°, 15° from the given directions at each site?

In L260 and following, an attempt has already been made to explain that the concept of the effective wind is the orthogonal projection of the wind vector to the "most relevant" wind direction for each location. This „most relevant" wind direction is the wind direction that caused the highest water level at the specific location. Each wind vector is then projected to this direction and the resulting length of the wind vector is used as the 'effective wind speed'. So no directional range is used to access the effective wind speed, rather each wind speed is used with its corresponding attribution in the „most relevant" direction.
However, for a better readability we shift the definition of the effective wind in Section 2

*„2.3.3 Effective wind*

*Wind statistics in the south-western Baltic Sea are shifted towards westerly components, especially for strong winds. However, strong westerly winds do not generate high water levels in the south-western Baltic Sea, so using all wind directions without modification may lead to biased statistical measures of extreme water levels. We are, therefore, only interested in the strong winds that are responsible for the generation of extreme water levels, i.e. winds coming from a particular direction depending on the orientation of the coast.*
*In order to calculate return values only for such wind components, the concept of the effective wind (Ganske et al., 2018) has been used. The effective wind is an orthogonal projection of the wind vector onto the prevailing surge-generating wind direction. Thus, instead of using only wind events within a given directional range, each wind event with its orthogonal portion of wind speed relative to a defined direction is used for statistical analysis. Here, for simplicity, we used as a reference direction the wind direction that occurred during the highest water level event during the hindcast period at each of the three sites. Using this approach, the effective wind direction is northeast (42.6 °) at Travemünde, north (354.6 °) at Warnemünde and east (90.6 °) at Flensburg. With these wind directions the effective wind is derived and used to estimate the return values for each location."*

L260 to 268 will be then be shortened to

*„The extent to which the wind situation was exceptional for the three events in 2017, 2019 and 2023 is analysed using the concept of the effective wind."*

Fig 7. Would be good to add effective wind speed for each panel or to the figure caption.

We will change the label to effective wind. In the figure caption it is already described as effective wind speed.

Line 310: Interesting to notice that nearly identical wind speeds and directions at Flensburg have led to different water levels in 2023 and 1958-1979. Was the prefilling component very small during the previous event in period 1958-1979 (yellow marker, Fig. 9)?

Actually this event is already mentioned between L 305 and L307

Fig. 9. please specify x-axis label 'ff', that should mean wind speed?

„ff" will be change to „wind speed"